# Clinical experience of genome-wide non-invasive prenatal testing as a first-tier screening test in a cohort of 59,771 pregnancies

Jianxin Zhen[1]❦*, Liting Zhang[1]❦, Huilin Wang[1], Xi Chen[1], Weihong Wang[1], Lili Li[2], Quanfu Zhang[1,2]*

1 Central Laboratory, Shenzhen Baoan Women's and Children's Hospital, Shenzhen, Guangdong Province, China, 2 Department of Obstetrics, Shenzhen Baoan Women's and Children's Hospital, Shenzhen, Guangdong Province, China

❦ These authors contributed equally to this work.
* jxzhen@qq.com (JZ); szzhqf@126.com (QZ)

## Abstract

### Objective

Genome-wide non-invasive prenatal testing (GW-NIPT) for prenatal screening has been widely implemented. However, the related clinical data is still insufficient. Here, we evaluated the clinical performance of GW-NIPT as a first-tier screening test for detecting fetal aneuploidy and copy number variation (CNV).

### Methods

The study included 59,877 pregnant women who underwent GW-NIPT at Shenzhen Baoan Women's and Children's Hospital, China, from November 2017 to May 2021. NIPT was performed on the BGISEQ-500 platform. Fetal karyotype analysis, chromosomal microarray analysis (CMA) and fluorescence in situ hybridization were used for invasive diagnostic procedures, and postnatal outcomes were collected.

### Results

Among 59,877 pregnant women who underwent GW-NIPT, 59,771 were successfully tested. Of these, 499 (0.83%) were identified with 504 high-risk fetal chromosomal abnormalities, including 5 cases each carrying two distinct abnormalities. Follow-up analysis demonstrated that GW-NIPT sensitivity exceeded 97% for fetal aneuploidies and was 63.6% for CNV (≥5Mb). The positive predictive values for T21, T18, T13, sex chromosome aneuploidy, rare autosomal aneuploidy, and CNV (≥5Mb) were calculated as 83.1%, 25.8%, 10.3%, 51.9%, 2.0%, and 33.9%, respectively. For confirmed fetal mosaicism, the detection rate of NIPT was 70.6%, which was consistent with that of CMA (70.6%).

**Data availability statement:** All relevant data are within the manuscript and its Supporting information files.

**Funding:** This research was supported by 2024 High-quality Development Research Project of Shenzhen Bao'an Public Hospital (Grant number: BAGZL2024154), and National Natural Science Foundation of China (Grant number: 82203291). The funders had no role in study design, data collection and analysis, decision to publish, or preparation of the manuscript.

**Competing interests:** The authors have declared that no competing interests exist.

## Conclusions

GW-NIPT has high sensitivity in screening fetal aneuploidy and moderate clinical utility in detecting CNV and fetal mosaicism, demonstrating that GW-NIPT holds significant application value in current and future prenatal screening procedures.

## Introduction

The emergence of non-invasive prenatal testing (NIPT) for cell-free fetal DNA (cffDNA) in maternal peripheral blood has greatly improved the convenience of prenatal testing for women, avoiding some pregnant women from directly undergoing invasive procedures that may increase the risk of miscarriage [1,2]. Since the discovery of cffDNA in maternal blood in 1997 [3], noninvasive prenatal diagnosis has been developed and introduced into clinical practice in 2011 [4]. At present, NIPT is becoming increasingly mature and widely used for antenatal screening of fetal trisomy 21 (Down syndrome), trisomy 18 (Edwards syndrome) and trisomy 13 (Patau syndrome). In addition, the additional findings, comprising fetal sex chromosome aneuploidy (SCA), rare autosomal aneuploidy (RAA) and copy number variation (CNV), can be detected by NIPT based on whole-genome sequencing [1,5–7].

Previous studies reported that NIPT had high sensitivities and low misdiagnosis rates for T21, T18 and T13 [5–8]. Introducing NIPT after combined first-trimester screening can significantly increased the screening specificity of T21, T18 and T13 at no loss of sensitivity, that will greatly reduce the number of invasive tests [9]. For additional finding of NIPT, SCA commonly includes 45, X (Turner syndrome), 47, XXY (Klinefelter syndrome), 47, XXX, and 47, XYY (Jacob's syndrome). Compared with monomer X, it has a higher positive predictive value (PPV) in predicting trisomy of sex chromosomes [10,11]. However, the detection performance of NIPT for RAA is not good enough, and van Prooyen Schuurman L et al. found that 91.3% of rare autosomal trisomy (RAT) originated from confined placental mosaicism (CPM) [12]. As for CNV, the accuracy of NIPT may vary according to the mutated fragments or classic microdeletion/microduplication syndrome (MMS) [13,14].

Fetal/placental mosaicisms are the major factors contributing to false-positive and false-negative results of NIPT [1,8]. CffDNA originates from placental trophoblast cells, which facilitates the detection of CPM, and further invasive testing is needed to distinguish between CPM and fetal mosaicisms. Both CPM and fetal mosaicisms have adverse effects on fetal growth. CPM cases are significantly associated with adverse pregnancy outcomes such as fetal growth restriction, pre-eclampsia, and low birth weight [12,15]. In fetal mosaicisms, unbalanced chromosome anomalies may pose a risk of developmental disorders and potential mental handicap, leading to termination of pregnancy [16].

American College of Medical Genetics and Genomics (ACMG) strongly recommends NIPT for all pregnant women with singleton and twin gestations for fetal common trisomies and SCA [17]. In 2017, the prenatal high-throughput genetic

screening projects for T21, T18 and T13 have been subsumed in the public welfare projects in Shenzhen City, China, facilitating the uptake of NIPT among local pregnant women. However, clinical research on genome-wide non-invasive prenatal testing (GW-NIPT) remains insufficient, with limited evidence available to establish clear clinical guidelines for additional findings beyond common trisomies [6,17,18]. And the studies on additional findings of NIPT often relies on increased sequencing depth for analysis [13,14,19,20], which expands the cost of testing. Besides, there are few reports on GW-NIPT detection of fetal mosaicisms, and more evidence is needed to explore the performance of NIPT screening for fetal mosaicisms.

In this study, we conducted a retrospective analysis on 59,771 pregnancies who underwent GW-NIPT, and evaluated the performance of the test as a first-tier prenatal screening method for common chromosomal trisomies and additional findings. We also analyzed the potential significance of the test in indicating fetal mosaicisms, aiming to further expand the clinical value of GW-NIPT in prenatal screening.

## Subjects and methods

### Study subjects

The retrospective study enrolled 59,877 singleton or twin pregnant women opting for GW-NIPT as a first-tier prenatal test from November 2017 to May 2021 at Shenzhen Baoan Women's and Children's Hospital in China. Among these, 59,771 women were included in the subsequent analysis due to 106 test failures. The exclusion criteria for NIPT were gestational age < 12 weeks, couples with clear chromosomal abnormality, women who had a malignancy during pregnancy and had received allogeneic blood transfusions, transplantation surgery, or allogeneic cell therapy within 1 year. Before the test, all participants received clinical counselling, including the content, requirements, advantages, and limitations of the test. The results of NIPT were interpreted by professional clinical doctors and provided patients with corresponding recommendations. All methods were performed according to the relevant guidelines or regulations.

### Ethics statement

The study was approved by the Ethics Committee of Shenzhen Baoan Women's and Children's Hospital (LLSC-2021-04-01-10-KS). Written informed consent was obtained from individual participants or their guardians. Records were accessed on 20th March 2022 and subsequently anonymized to protect the personal privacy of all pregnant women.

### Laboratory analysis of GW-NIPT

5mL of maternal peripheral blood was collected in an EDTA tube and kept standing at room temperature for 30 minutes. Plasma was extracted from each blood sample using the following two-step centrifugation protocol. Firstly, the whole-blood specimens were centrifuged for 10 minutes at 1600 g at 4°C to separate plasma from blood cells. Subsequently, the supernatant was transferred into a new tube and centrifuged at 16000 g for 10 minutes at 4°C to precipitate the remaining cells. The prepared plasma was stored at − 70°C.

DNA extraction, concentration detection, and DNA library construction were performed on plasma samples using fetal chromosome aneuploidy detection kit package (BGI, Shenzhen, China) in the method of combinatorial probe-anchor synthesis. After libraries pooling, single stranded formation and cyclization of the samples, DNA Nanoball (DNB) was prepared and quantified using Qubit ssDNA Assay kits (Thermo Fisher, Waltham, USA). DNB concentrations in the range of 8−40 ng/μl was considered qualified. Sequencing was performed on the BGISEQ-500 platform (BGI, Shenzhen, China) with average sequencing depth of 0.17 × . These sequencing data were analyzed using BGI's automated analysis and interpretation platform, BGI HALOS. The minimum detectable fraction of cffDNA was 3.5%. Based on the Z-score (normal range, −3 < Z < 3), all chromosomes were evaluated to determine the fetal aneuploidy status.

## Invasive prenatal diagnosis and postnatal outcomes

Pregnant women who required further examination would have amniocentesis for karyotyping, chromosomal microarray analysis (CMA), or fluorescence in situ hybridization (FISH), following the results of NIPT, ultrasound examination and other medical history. A small number of cases underwent diagnostic testing using chorionic villus sampling (CVS), umbilical cord blood, and fetal tissue.

30ml of amniotic fluid was obtained between 16~24 weeks of pregnancy for karyotype testing and CMA. After standard metaphase conversion of cultured fetal cells, chromosome analysis was performed on at least 20 metaphase cells of each specimen, while each mosaic specimen was counted in more than 30 metaphase cells. It was also confirmed that mosaicism were detected in both two bottles of cultured cells. The G-bands level in this laboratory was 320~400.

CMA was conducted using comparative genomic hybridization (CGH) technology. QIAamp® DNA Blood Mini Kit (QIAGEN, Hilden, Germany) was used to extract DNA from samples. Then, the samples were subjected to thermal cracking, purification, and hybridization using the Oligo CGH Microarray Kit (Agilent, California, USA) with Fetal DNA Chip Version 1.2. Finally, SureScan Microarray Scanner (Agilent, California, USA) was used for scanning the result. The Derivative Log Ratio Standard Deviation was ensured to be less than 0.2. Data analysis was referred to the human genome hg19 reference sequence and the pathogenic significance of CNV was evaluated according to the ACMG guidelines [21]. Based on the evaluation of the pathogenicity of genetic variants referring to Population, Disease-Specific, and Sequence Databases, as well as published literature, CNVs were categorized into two groups: "pathogenic/likely pathogenic (P/LP)" and "variant of uncertain significance (VUS)". FISH experiments were conducted using the FISH detection kit (Jinpujia, Beijing, China).

For postnatal outcomes, we included information on the types of delivery (live birth vs. stillbirth), the presence of any birth defects, and the clinical evaluation of birth defects. When postnatal outcomes was not recorded, potential reasons included termination of pregnancy (TOP) or delivery at another hospital, which we did not differentiate in this study.

We classified NIPT results into true positive (TP), false positive (FP), true negative (TN) and false negative (FN) according to invasive testing results or the postnatal outcomes. The criteria for judgment of NIPT results are as follows: (1) T21, T18, T13 and RAA can be comfirmed via invasive testing (karyotyping, CMA or FISH) or postnatal outcomes due to their distinct clinical features, such as atelencephalia in common trisomies and cardiovascular defects in RAA [22,23]. (2) SCA was comfirmed by invasive testing of karyotyping, CMA or FISH. (3) CNV (≥5Mb) was comfirmed by invasive testing of CMA.

## Statistical analysis

IBM SPSS statistics 27.0.1 (SPSS Inc., Chicago, IL, USA) was used to analyse the data. Continuous variables were presented in the form of mean ± standard deviation, median, or range for uptake, while categorical variables were expressed as proportions. Sensitivity, specificity and PPV were calculated using 2 × 2 tables with following formula: sensitivity = TP/(TP + FN), specificity = TN/(TN + FP) and PPV = TP/(TP + FP). On the bases of a standard normal distribution, 95% confidence intervals (CI) were calculated by Normal Approximation method and Wilson method.

## Results

### Study population characteristics and GW-NIPT results

Demographic characteristics of the study population were summarized in Table 1. The average maternal age was 29.5 (±4.1) years which is slightly higher than the national average age of pregnant women of 28.8 (±5.1) years from 2016 to 2020 [24], and 11.6% (6946/59771) of pregnant women were in the advanced maternal age (≥35). The mean body mass index (BMI) was 21.7 (±3.0) kg/m$^2$ and the median gestational age was 17 weeks (range: 12–39.3).

A total of 59,877 pregnancies who underwent GW-NIPT from November 2017 to May 2021 were accessioned for this study and 106 were tested failure (fetal fraction <3.5%). The test failure rate was 0.18% (106/59877) and 59,771 cases

**Table 1. Demographic characteristics of the the study cohort.**

| Maternal characteristics | n (%) |
|---|---|
| *Maternal age (years)* | |
| <35 | 52825 (88.4) |
| ≥35 | 6946 (11.6) |
| *BMI (kg/m²)* | |
| <18.5 | 6469 (10.8) |
| 18.5 - 23.9 | 41179 (68.9) |
| 24.0 - 27.9 | 10001 (16.7) |
| ≥28.0 | 2109 (3.5) |
| Unrecorded | 13 (0.0) |
| *Gestation age (weeks)* | |
| <28 | 58928 (98.6) |
| ≥28 | 835 (1.4) |
| Unrecorded | 8 (0.0) |
| *Pregnancy types* | |
| Singleton pregnancy | 58342 (97.6) |
| Twin pregnancy | 1421 (2.4) |
| Unrecorded | 8 (0.0) |

n, number; BMI, body mass index.

were enrolled in further analysis (Fig 1). In the results of GW-NIPT, 499 (0.8%) patients were suspected to be high risk, of which 5 carried two types of abnormal high risk. Among 504 high-risk cases, 367 cases were confirmed by follow-up diagnosis, and 137 cases were lost to follow-up. 59,272 cases showed low-risk results for all of the detected chromosomal abnormalities, of which 1,402 cases underwent invasive CMA and 1,349 cases conducted invasive karyotype analysis based on other clinical evaluations, including 1,315 cases that underwent both CMA and karyotyping. Among the remaining low-risk cases of NIPT, 43,413 have recorded postnatal outcomes and 14,423 cases lost to follow-up.

## Performance of GW-NIPT for screening common trisomies

In the 89 cases of trisomy 21 high risk of NIPT, 24 cases were lost without confirmation and 65 cases were followed up by cytogenetic testing and clinical diagnosis (Table 2). Among them, NIPT results were confrmed in 54 cases, including 53 were confrmed by cytogenetic testing using amniotic fluid or fetal tissue and one was identified by clinical diagnosis after intrauterine fetal demise. Eleven cases were false positives of NIPT, comprising six were confirmed by cytogenetic testing of amniocytes and five healthy neonates without cytogenetic confirmation (S1 Table). Among the 59,684 cases with low risk of T21 indicated by NIPT, one case of false negative of NIPT was confirmed as trisomy 21 mosaic (47,XN,+21[5]/46,XN[95]) by karyotyping of amniocytes (S2 Table), and 45,208 cases were comfirmed as true negative of NIPT. In summary, the sensitivity, specificity and PPV of NIPT for trisomy 21 were 98.2%, 100% and 83.1%, respectively (Table 2).

For trisomy 18, 36 cases were considered positive by NIPT. In these cases, five were missing and 31 had undergone cytogenetic testing and/or clinical diagnosis. NIPT results were confrmed in eight cases by cytogenetic testing in amniotic fluid, of which one case was confirmed as mosaic trisomy 18 (47,XN,+18[21]/46,XN[79]). 23 cases were discordant positives of NIPT, including 19 were detected by cytogenetic testing using amniotic fluid or cord blood and four were healthy births without cytogenetic confirmation (S1 Table). In cases where NIPT indicated low risk of T18, no false negative case

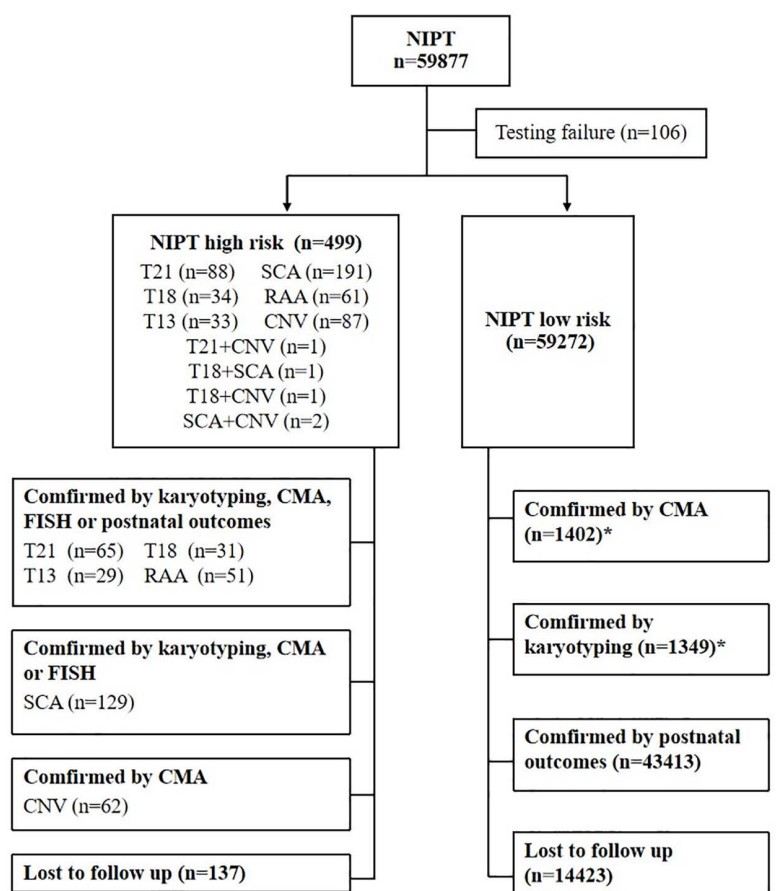

**Fig 1. Flowchart of the NIPT results and outcomes of pregnant women.** NIPT, non-invasive prenatal test; n, number; T, trisomy; SCA, sex chromosome aneuploidy; RAA, rare autosomal aneuploidy; CNV, copy number variation; CMA, chromosomal microarray analysis; FISH, fluorescence in situ hybridization. *Include 1,315 cases that underwent both karyotyping and CMA.

**Table 2. NIPT performance for common trisomies, SCA, RAA and CNV.**

| Abnormal types | NIPT high risk | | | | NIPT low risk | | | | Sensitivity, % (95% CI) | Specificity, % (95% CI) | PPV, % (95% CI) |
|---|---|---|---|---|---|---|---|---|---|---|---|
| | n | lost | TP | FP | n | lost | FN | TN | | | |
| T21 | 89 | 24 | 54 | 11 | 59684 | 14475 | 1* | 45208 | 98.2 (94.6-100) | 100 (99.9-100) | 83.1 (74.0-92.2) |
| T18 | 36 | 5 | 8 | 23 | 59735 | 14494 | 0 | 45243 | 100 (67.6-100) | 99.9 (99.9-100) | 25.8 (10.4-41.2) |
| T13 | 33 | 4 | 3 | 26 | 59738 | 14495 | 0 | 45245 | 100 (43.8-100) | 99.9 (99.9-100) | 10.3 (0-21.4) |
| SCA | 194 | 65 | 67 | 62 | 59577 | 57942 | 2$ | 1633 | 97.1 (93.1-100) | 96.3 (95.4-97.2) | 51.9 (43.3-60.6) |
| RAA | 61 | 10 | 1 | 50 | 59710 | 14489 | 0 | 45223 | 100 (20.6-100) | 99.9 (99.8-99.9) | 2.0 (0-5.8) |
| CNV | 91 | 29 | 21 | 41 | 59680 | 58032 | 12 | 1636 | 63.6 (47.2-80.0) | 97.6 (96.8-98.3) | 33.9 (22.1-45.6) |

NIPT, non-invasive prenatal test; n, number; TP, true positive; FP, false positive; FN, false negative; TN, ture negative; CI, confidence interval; PPV, positive predictive value; T, trisomy; SCA, sex chromosome aneuploidy; RAA, rare autosomal aneuploid; CNV, copy number variation. * Confirmed as trisomy 21 mosaicism. $Confirmed as SCA mosaicisms.

and 45,245 ture negative cases were reported. Thus, the sensitivity, specificity and PPV of NIPT for trisomy 18 were 100%, 99.9% and 25.8%, respectively (Table 2).

As for trisomy 13, 33 cases were identified by NIPT as high-risk cases. Four cases were lost without confirmation and 29 cases were followed up by cytogenetic testing and/or clinical diagnosis. For three of the 29 cases, the NIPT result was confirmed by cytogenetic testing in amniocytes. The remaining 26 cases were false positives of NIPT, which involved 18 were confirmed as normal by cytogenetic testing of amniocytes or cord blood and eight were healthy births without cytogenetic confirmation (S1 Table). In summary, the sensitivity, specificity and PPV of NIPT for trisomy 13 were 100%, 99.9% and 10.3%, respectively (Table 2).

## Performance of GW-NIPT for screening SCAs, RAAs and CNVs (≥5 Mb)

A total of SCA detected by NIPT were 194 cases (Table 2). The results were confrmed in 67 cases by cytogenetic testing using amniotic fluid or cord blood, including ten confirmed as mosaic SCA and two diagnosed in other hospitals as XXY and XXX respectively. In 62 cases of discordant positive of NIPT, one case was XXY high risk but confirmed as XXYY by karyotyping with amniocytes (S3 Table). Of the 129 diagnosed cases, 17 were reported as XXX by NIPT with 10 (58.8%) true positives, 33 were detected as XXY with 30 (90.9%) true positives, 18 were detected as XYY by NIPT with three discordant positives (83.3%) and 61 were reported as X with 12 (19.7%) true positives (Fig 2). Among 59,577 cases of SCA low risk of NIPT, 1,635 underwent invasive testing, of which two cases were false negative of NIPT and confirmed as mosaic SCA by karyotyping with amniocytes (S2 Table). In summary, the overall sensitivity, specificity and PPV of NIPT for SCA were 97.1%, 96.3% and 51.9%, respectively (Table 2).

For RAA, 61 cases were considered positive by NIPT. Of these, 10 cases were lost and 51 cases had cytogenetic testing and/or pregnancy outcome (Table 2, S4 Table). Ture positive were confirmed in one case with 47,XN,+2[7]/46,XN[93] by karyotyping. 50 cases were confirmed as false positives of NIPT, which included 34 cases with diagnostic test in amniotic fluid or cord blood and 16 healthy births without cytogenetic confirmation. No false negative cases were reported. As a result, the sensitivity, specificity and PPV for RAA was 100%, 99.9% and 2.0%, respectively.

Finally, 91 cases were identified by NIPT as CNV (≥5 Mb) high-risk with 29 missing cases and 62 cases diagnosed by CMA (Table 2, S5 Table). In the 62 cases, 21 were confirmed to be consistent with NIPT results, including 3 MMSs (S5 Table), and 41 were confirmed as discordant positive using amniotic fluid or cord blood. Among a total of 1,648 low-risk cases of CNV (≥5 Mb) conducted follow-up diagnosis, 12 cases were comfirmed as false negative of NIPT by CMA of amniotic fluid, cord blood or fetal tissue (S2 Table). Thus, the sensitivity, specificity and PPV of NIPT for CNV were 63.6%, 97.6% and 33.9%, respectively. Furthermore, we classified CNV into three categories according to abnormal fragment

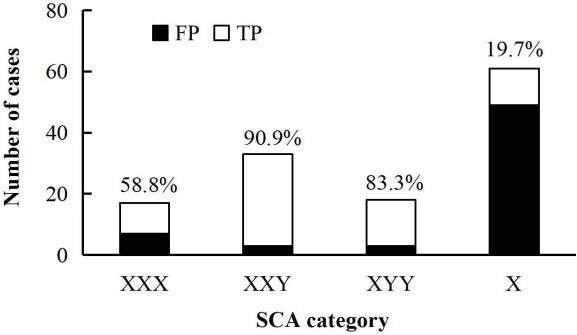

**Fig 2. Distribution of diagnosed cases of SCA high risk of NIPT.** The number at the top of the pillar represents the PPV for each SCA category. NIPT, non-invasive prenatal test; FP, false positive; TP, true positive; SCA, sex chromosome aneuploidy.

sizes: 5–10 Mb, 10–20 Mb and >20 Mb. Of the 62 diagnosed cases, 30 were in 5–10 Mb group comprising 13 (43.3%) cases of true positive, 17 cases were in 10–20 Mb group with six (35.3%) true positives and 15 were in >20 Mb group with two (13.3%) true-positive cases (Fig 3). Regarding comfirmed CNV, more adverse postnatal outcomes occurred in cases with P/LP variants compared to cases with VUS variants, including premature birth, stillbirth or possible TOP (S2 and S5 Table).

## Fetal mosaicism

Through amniotic fluid karyotype analysis, a total of 17 cases were diagnosed as mosaics (Table 3), among which 12 cases (70.6%) were consistently identified by NIPT and 12 cases (70.6%) were detected by CMA. Two cases were identified as low-risk by NIPT, while karyotype analysis revealed SCA mosaic and chromosome 18 deletion mosaicism, which were consistent with the CMA results. One case indicating trisomy 2 by NIPT was consistent in both karyotype and CMA results. One case with high-risk trisomy 8 detected by NIPT showed a karyotype of 47,XN,+21[5]/46,XN[95], while its CMA result was negative. In addition, NIPT identified two cases with high-risk trisomy 18. One was diagnosed by karyotype analysis as 47,XN,+18[21]/46,XN[79], which was consistent with the CMA result. The other one was 47,XYY[6]/46,XY[94] diagnosed by karyotype, but its result of CMA was negative. Nine cases with high-risk monosomy X and one case with XXX identified by NIPT were all confirmed by karyotype analysis, however, three cases were not detected by CMA. The last case had a karyotype result of 46,XN,t(5;7)(p10;p10)[13]/46,XN[87], indicating an isochromosome translocation between the short arms of chromosomes 5 and 7, which is beyond the detection range of NIPT and CMA. Thus, the CMA result was normal, meaning that it was consistent with the result of karyotyping, and the NIPT finding of a microdeletion in chromosome 15 was a false positive result.

## Discussion

NIPT is a screening test in the second trimester, detecting cffDNA in maternal peripheral blood. Compared with traditional maternal serum screening, NIPT has higher sensitivity and specificity [1,17]. At present, more than 27 countries have implemented NIPT, reducing the proportion of women who choose invasive prenatal diagnosis after high-risk biochemical screening from 75% to 43% [25], optimizing the prenatal testing process and better preventing disabilities. More clinical data is still needed to evaluate the screening performance of NIPT for additional findings beyond common chromosomal trisomy. In this study, we investigated 59,771 pregnant women who underwent genome-wide NIPT to evaluate the screening performance of the test for common chromosomal trisomy, SCA, RAA, CNV (≥5 Mb) and fetal mosaicism.

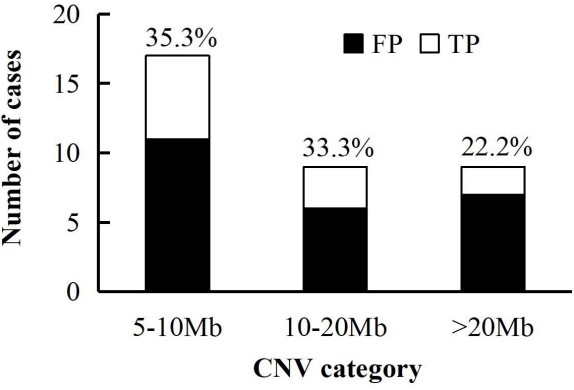

**Fig 3. Distribution of diagnosed cases of CNV high risk of NIPT.** The number at the top of the pillar represents the PPV for each CNV category. NIPT, non-invasive prenatal test; FP, false positive; TP, true positive; CNV, copy number variation; Mb, megabyte.

**Table 3. Overview of fetal mosaicisms.**

| NIPT result | MA (y) | GA (w) | BMI | Karyotyping (in AF) | CMA (in AF) | Pregnancy outcome |
|---|---|---|---|---|---|---|
| low-risk | 39 | 12 + 6 | 24.0 | 45,X[58]/46,X,idic(X)(p11.2)[42] | Xp22.33p11.22 × 1 | Premature birth and stillbirth |
| low-risk | 36 | 20 + 5 | 20.0 | 46,XN,del(18)(q21.3)[19]/46,XN[81] | 18q21.31q23 ×1~2 | |
| T2 | 34 | 12 + 4 | 21.1 | 47,XN, + 2[7]/46,XN[93] | 2p25.3q37.3 ×2~3 | |
| T8 | 29 | 12 + 5 | 19.2 | 47,XN, + 21[5]/46,XN[95] | Normal | |
| T18 | 29 | 12 + 4 | 18.1 | 47,XN, + 18[21]/46,XN[79] | 18 p11.32-q23 ×2~3 | |
| T18 | 32 | 13 + 3 | 20.1 | 47,XYY[6]/46,XY[94] | Normal | Healthy live born |
| 45, X | 26 | 12 + 3 | 17.7 | 45,X[8]/46,XX[92] | Normal | Live born with congenital anomalies |
| 45, X | 30 | 13 + 4 | 21.5 | 45,X[12]/46,XX[88] | X p22.33-q28 ×1~2 | |
| 45, X | 25 | 13 + 2 | 17.9 | 45,X[12]/46,XX[88] | Normal | Healthy live born |
| 45, X | 31 | 13 + 3 | 18.0 | 45,X[60]/47,XXX[40] | X p22.33-q28 ×1~2 | |
| 45, X | 31 | 13 | 23.5 | 45,X[8]/46,XX[92] | Normal | |
| 45, X | 35 | 14 + 1 | 22.7 | 45,X[93]/46,X,idic(X)(p11.2)[7] | X p22.33-q28 ×1 | |
| 45, X | 30 | 13 + 3 | 15.8 | 45,X[67]/47,XXX[33] | X p22.33-q28 ×1~2 | |
| 45, X | 30 | 12 + 0 | 19.6 | 45,X[10]/46,XX[90] | Xp22.33q28 ×1~2 | |
| 45, X | 30 | 17 + 6 | 22.6 | 45,X,16qh+[55]/46,X,mar,16qh+[45] | Xp22.33p11.4 × 1 | |
| 47, XXX | 28 | 23 + 2 | 22.8 | 47,XXX[75]/46,XX[25] | X p22.3-q28 ×3 | |
| Del 15 q11.2-q13.3 | 27 | 12 + 2 | 19.6 | 46,XN,t(5;7)(p10;p10)[13]/46,XN[87] | Normal | Healthy live born |

NIPT, non-invasive prenatal test; MA, maternal age (years); GA, gestational age (weeks); BMI, body mass index; AF, amniotic fluid; CMA, chromosomal microarray analysis; T, trisomy; del, deletion; Mb, megabyte.

We used next-generation sequencing technology for NIPT. In the research cohort, a total of 499 patients (0.83%) were indicated to have a high risk of chromosomal abnormalities. This result is similar to previous studies that found a positive rate of 0.75% (without SCA) for NIPT screening in the general obstetric population [5], while the positive rate of NIPT screening was 2.0% in the women with high-risk and intermediate-risk results of traditional combined first-trimester screening (cFTS), including blood screening combined with an ultrasound examination [7]. Compared with traditional cFTS, NIPT has higher reliability. The follow-up diagnosis confirmed that there were 213 (0.4%) false positive cases of T21, T18, T13 and additional findings of NIPT, however, the false positive rate of traditional cFTS for common chromosomal trisomies was 4.7% [26].

Follow-up diagnostic testing confirmed that NIPT demonstrated high sensitivity for T21, T18 and T13, with respective values of 98.2%, 100%, and 100%. Only one false negative case was confirmed as trisomy 21 mosaicism, while NIPT indicated trisomy 8 high risk. This aligned with a meta-analysis of previous studies reporting high sensitivity for common chromosomal trisomies in fetus, with T21 at 98.8%, T18 at 98.8%, and T13 at 100% [27]. In this study, PPVs of NIPT for T21, T18 and T13 were 83.1%, 25.8%, and 10.3%, respectively, highlight the need for further diagnostic confirmation following a positive NIPT result. The PPVs reported by van der Meij KRM et al. of NIPT for T21, T18 and T13 were 96%,

98%, and 53%, respectively [5], which were higher than our results. However, the corresponding sensitivities in their study were slightly lower, at 98%, 91% and 100%, respectively, resulting in more false-negative cases. These variations highlighted that the balance between PPVs and sensitivities could be influenced by the specific NIPT platforms used. In the other hand, several factors can also affect these two parameters, such as sample size, fetal or placental mosaicism, the 'vanishing twin' phenomenon, and maternal copy number variations [1,8,28]. For instance, CPM can lead to false-positive results, as the cffDNA may originate from the placenta rather than the fetus [28]. Additionally, maternal copy number variations can affect the accuracy of NIPT results by introducing additional genetic signals that may be misinterpreted as fetal abnormalities.

In this study, the sensitivity and PPV of SCA detected by NIPT were 97.1% and 51.9%, respectively. Kim H et al. reported comparable NIPT PPV for SCA (51.4% overall), with subtype-specific values of 88.9% (47,XXX), 71.4% (47,XXY), 60.0% (47,XYY), and 18.8% (45,X) [11]. Our results demonstrated similar trends: 58.8% (47,XXX), 90.9% (47,XXY), 83.3% (47,XYY), and 19.7% (45,X), indicating that NIPT performed better in predicting sex chromosome trisomies compared with monosomy X [10,11,13,19]. The reason for the low accuracy of NIPT in predicting monomer X might be: (i) CPM, as mentioned above; (ii) The sequencing length of NIPT is 35 bases, and the pseudoautosomal regions at both ends of the X and Y chromosomes are highly homologous, making it prone to read error during sequencing these locations [10,24]; (iii) Low content of GC bases on the X chromosome leads to increased variability in amplification efficiency and difficulty in detection [29]; (iv) As the mother's age increases, the chromosomal karyotype of some female cells changes from XX to X0/XX mosaicism [30]. In additon, we found a case which was detected as XXY high risk by NIPT, was revealed as a 48, XXYY karyotype through the analysis of amniocytes. Thus, combining NIPT screening with prenatal diagnosis is beneficial for the discovery and diagnosis of rare fetal chromosomal abnormalities.

We also analyzed the high-risk cases of RAA of NIPT, with the sensitivity and PPV of 100% and 2.0%, respectively. In previous large-scale obstetric population studies, the incidence rates of RAT were 0.014% (15/110739) [12] and 0.015% (14/94085) [13], and the corresponding PPVs were 7.6% and 28.6% respectively. However, there was only one case of RAT in our cohort, which indicated its incidence rate was 0.0017% (1/59771). Routine follow-up on pregnancy outcomes of RAA high risk of NIPT were conducted, including 39 live-born cases with no significant abnormalities at birth, 2 cases with congenital disease, 3 stillbirths and the rest without registration. The outcomes of RAA of fetus are mostly early miscarriage and death, expecially in trisomy 16, 22, and 15 fetus [31,32]. Thus, even without diagnostic testing, cases with normal birth can exclude RAA. Fetus who borned with congenital disease did not have diagnostic testing. However, all of three stillbirth cases had negative results of diagnostic test in amniotic fluid, and the cause of death may be due to placental mosaicism, malnutrition during pregnancy or premature birth. Therefore, if conditions permit, cases with high risk of RAA of NIPT should be performed ultrasound, to monitor fetal growth and avoid potentially structural abnormalities in fetus and unnecessary invasive testing [5,33].

The overall sensitivity of NIPT in detecting high risk CNV (≥5 Mb) was 63.6%. Previous studies reported that the sensitivities of NIPT screening for CNV were >90% [13,20,34], and the main reason is that generally CNVs have inapparent clinical manifestations in fetus, resulting in some of potential false negatives of NIPT without diagnosis. In this study, patients with low-risk CNV (≥5 Mb) of NIPT would take diagnostic tests based on abnormal ultrasound results, medical history or personal intention. Besides, the PPV of NIPT for CNV (≥5 Mb) was 33.9%, meaning that NIPT had a certain screening advantage for chromosomal fragment abnormalities. Other studies reported an overall PPV ranging from 38.1% to 50% for CNV by NIPT-Plus with deeper sequencing depth than ours [13,14,19,20]. Schwartz S et al. reported an PPV of 9.2% in 349 patients screened for 1p, 4p, 5p, 15q, or 22q microdeletions of NIPT, and their research data were obtained from seven different laboratories [35].

Additionally, among 91 cases with high-risk CNV (≥5 Mb) indications by NIPT, we identified 17 cases of MMSs. Among these cases, three were confirmed as true positives, including one case each of Wolf-Hirschhorn syndrome, 18q deletion syndrome, and Cri-du-Chat syndrome (CDC). This indicates that GW-NIPT is capable of detecting

pathogenic MMS, albeit with a low true-positive rate. Studies report variable PPVs of NIPT for specific MMS.For example, Xue H et al. found PPVs of 50%, 50% and 0% for CDC, PWS/AS and 1p36 deletion, respectively [14] and Petersen et al. reported corresponding PPVs of 0%, 0% and 14% [36].These may be influenced by factors such as CNVs carried by pregnant woman, sequencing coverage, bioinformatics algorithms, fetal DNA fraction, etc. [37]. Furthermore, our results demonstrate that fetuses with P/LP CNVs were more likely to result in TOP or adverse postnatal outcomes compared to those with VUS CNVs. Besides, long-term follow-up and monitoring should be implemented for cases with high-risk CNVs who continue to live birth.

17 cases of fetal mosaicisms were confirmed by karyotyping of amniocytes, which both underwent NIPT and CMA. We found that the detection rate of fetal mosaicism by NIPT was 70.6% (12/17), which was consistent with that of CMA (70.6%, 12/17). At present, researches on the detection of fetal mosaicisms are mainly focused on karyotype analysis and CMA [16,38,39], both of which have their own advantages and disadvantages. The former mainly evaluates mosaicisms with balanced translocation or mosaic with constant total gene dosage, while the latter has more advantages in detecting mosaicisms with short fragment of chromosomal abnormalities [39]. Our results revealed that NIPT had certain detection efficacy for fetal mosaics, especially in mosaicism with chromosomal aneuploidy.

Based on targeted array CGH using Fetal DNA Chip Version 1.2, CMA can detect all common aneuploidy diseases and over 100 types of MMSs of known gene sequence, yet has limitations like missing some small chromosomal abnormalities, copy number polymorphisms, mosaicism, and rearrangements. GW-NIPT enables detection of a broader range of genomic variations and is capable of identifying atypical, rare, or subchromosomal CNVs that would not be identified by targeted assays. The Algorithms of GW-NIPT developed by BGI-Shenzhen were used to detect CNV and the reads were aligned to the human genome (hg19). After raw data processing, GC correction, intra-batch correction and Principal Component Analysis (PCA), Hidden Markov Model (HMM) was used to infer the hidden states of each window (normal, deletion and duplication) [40]. Concurrently, GW-NIPT identified 2 cases of SCA mosaicism (comfirmed by karyotyping) that were undetected by CMA. These findings underscore the clinical necessity of implementing GW-NIPT. This study has several limitations: (1) Incomplete postnatal outcomes, without distinguishing between TOP and lost to follow-up cases. (2) Long-term follow-up was not performed, and no data were collected on the long-term impact of NIPT results on neonatal outcomes or childhood health. (3) In-depth analysis on high-risk pregnancies—particularly those with maternal morbidity or negative neonatal outcomes—was insufficiently conducted. These methodological gaps should be prioritized in subsequent research to enhance the robustness of future studies.

## Conclusions

The clinical study of 59,771 pregnant women in the general obstetric population substantiated that GW-NIPT had high sensitivity in screening for fetal trisomy 21, 18, and 13 with PPV of 83.1%, 25.8%, and 10.3%, respectively. For additional findings, GW-NIPT showed good screening performance in detecting SCA with PPVs of 51.9%. The incidence rate of RAA was very low in this study, at 0.002% (1/59771), and the PPV of NIPT for RAA was 2%. For CNV, the overall PPV of NIPT was 33.9%. In addition, we found that NIPT had potential ability in screening fetal mosaicism. Compared to CMA, GW-NIPT has certain advantages in detecting mosaicism of fetal chromosomal aneuploidy. Based on the support of the policies, we suggest that GW-NIPT should be analyzed in conjunction with traditional serum screening, ultrasound testing and parents' medical history, which can optimize the prenatal testing process, as well as provide timely and comprehensive genetic counseling for patients.

## Supporting information

**S1 Table.  False-positive cases of trisomy 21, 18 and 13.**
(XLSX)

**S2 Table. False-negative cases of NIPT.**
(XLSX)

**S3 Table. False-positive cases of sex chromosome aneuploidies.**
(XLSX)

**S4 Table. Rare autosomal aneuploidies detected by NIPT.**
(XLSX)

**S5 Table. Copy number variations detected by NIPT.**
(XLSX)

## Acknowledgments

We would like to thank all the participants who contributed to the research.

## Author contributions

**Conceptualization:** Jianxin Zhen, Quanfu Zhang.

**Data curation:** Weihong Wang, Lili Li, Quanfu Zhang.

**Formal analysis:** Jianxin Zhen, Liting Zhang.

**Methodology:** Huilin Wang, Xi Chen.

**Writing – original draft:** Jianxin Zhen, Liting Zhang.

**Writing – review & editing:** Jianxin Zhen, Quanfu Zhang.

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
