## [Decision Letter · Decision Letter 0]

PONE-D-25-18695Clinical experience of genome-wide non-invasive prenatal testing as a first-tier screening test in a cohort of 59,771 pregnanciesPLOS ONE

Dear Dr. Zhen,

Thank you for submitting your manuscript to PLOS ONE. After careful consideration, we feel that it has merit but does not fully meet PLOS ONE’s publication criteria as it currently stands. Therefore, we invite you to submit a revised version of the manuscript that addresses the points raised during the review process.

We look forward to receiving your revised manuscript.

Kind regards,

Giuseppe Novelli

Academic Editor

PLOS ONE

Journal Requirements:

This research was supported by 2024 High-quality Development Research Project of Shenzhen Bao'an Public Hospital (Grant number: BAGZL2024154), and National Natural Science Foundation of China (Grant number: 82203291).

3. Please remove all personal information, ensure that the data shared are in accordance with participant consent, and re-upload a fully anonymized data set.

Reviewers' comments:

Reviewer's Responses to Questions

**Comments to the Author**

1. Is the manuscript technically sound, and do the data support the conclusions?

Reviewer #1: Yes

Reviewer #2: Partly

Reviewer #3: Yes

2. Has the statistical analysis been performed appropriately and rigorously? 

Reviewer #1: Yes

Reviewer #2: N/A

Reviewer #3: Yes

3. Have the authors made all data underlying the findings in their manuscript fully available?

Reviewer #1: Yes

Reviewer #2: Yes

Reviewer #3: Yes

4. Is the manuscript presented in an intelligible fashion and written in standard English?

Reviewer #1: Yes

Reviewer #2: Yes

Reviewer #3: Yes

5. Review Comments to the Author

Reviewer #1: This manuscript explores the use of genome-wide non-invasive prenatal testing (NIPT) for the detection of chromosomal aneuploidies and copy number variations (CNVs). The study addresses a timely and clinically relevant topic and presents data from a sizable cohort. Genome-wide NIPT has the potential to improve prenatal screening by expanding the range of detectable genomic alterations beyond common trisomies. However, several important aspects require clarification and revision before the manuscript is suitable for publication.

First, the manuscript does not report the test failure rate or the number of samples for which results were inconclusive or uninterpretable. This information is essential to assess the practical applicability and reliability of the test in a real-world clinical setting.

Second, while the authors report performance metrics such as sensitivity and specificity, the methodology behind these calculations is not sufficiently explained. It is unclear how true positives were defined and whether confirmation was based on invasive diagnostic testing, postnatal outcomes, or clinical evaluation. A detailed description of the reference standard and classification criteria is necessary to validate the reported performance.

Additionally, although the study employs a genome-wide approach, the discussion could better emphasize the advantages of this strategy. Specifically, the authors should elaborate on how genome-wide NIPT may increase diagnostic yield by detecting atypical, rare, or subchromosomal CNVs that would not be identified by targeted assays. This is a key strength of the approach and should be clearly articulated.

Importantly, the manuscript does not specify the resolution limit for CNV detection (i.e., the minimum size of CNVs that can be reliably identified). This technical parameter is critical for interpreting the scope and limitations of the assay and for comparing its performance with other technologies such as array-CGH. The authors should clearly indicate this resolution threshold and how it was validated.

Furthermore, a demographic summary of the study cohort—including maternal age, gestational age, and clinical indications for testing—would provide valuable context for the reader. The authors are also encouraged to include more detail on the bioinformatic workflow used for CNV calling, including the software tools, filtering criteria, and any validation methods applied. Sharing access to the analysis pipeline, when possible, would greatly enhance the study's transparency and reproducibility.

Finally, some minor editorial improvements could enhance the clarity and flow of the text, particularly in the abstract and results sections.

Reviewer #2: The study presents a total of 59,771 pregnancies who underwent NIPT from November 2017 to May 2021 where DNA extraction, concentration detection, and DNA library construction were performed on plasma samples using fetal chromosome aneuploidy detection kit package (BGI, Shenzhen, China) in the method of combinatorial probe-anchor synthesis.

The study of 59,771 pregnant women in the general obstetric population substantiated a low specificity for fetal trisomy 21, 18, and 13 with PPV of 83.1%, 25.8%, and 10.3%, respectively.

On the contrary the technical validation of the method (DOI: 10.1002/uog.16010) reported a high specificity with PPV of 99.98, 99.98, 99.9 for T21, T18, T13 respectively.

We note that the analysis of the study performance shows good sensitivity while the specificity is particularly low both with respect to the technical validation of the method and to international data.

This important difference raises a question of reliability that requires an answer. The authors compare the difference in specificity with an insufficient number of published case studies and suggest uncertain motivations. For example, the recent case study on approximately 72,000 consecutive cases from a single center in two years (DOI: 10.1002/pd.6580) performed with a different technique (no PCR, pair end sequencing) that reports PPV 99.3, 98.9, 82.22 for T21, T18, T13 respectively, was not considered.

Since low specificity is one of the limiting parameters in the cost-benefit evaluation of NIPT use leading to the execution of an improper number of invasive controls, the authors are asked for a complete review of the case studies and a reassessment of the conclusions regarding the reliability of the method used in the study.

Reviewer #3: The statistical data are generally comparable to those already reported in the existing literature. However, the manuscript does contribute by adding to the available dataset, particularly for rare autosomal aneuploidies (RAAs) and copy number variations (CNVs). The assessment for fetal mosaicism appears to be the most original and potentially valuable aspect of the study. The manuscript is complete and well presented. However, some points need to be reviewed:

• It is not entirely clear whether follow-up results were available for all cases identified as high- or low-risk by NIPT. Clarifying the extent and completeness of follow-up data.

• In some sections, the technical language is unclear. For example, in the cytogenetic analysis, 'karyotypes' are mentioned instead of 'metaphase' (in the paragraph titled Invasive prenatal diagnosis).

• In the classification of results, 'No Calls' are not mentioned.

• There is a lack of classification for the detected CNVs, distinguishing between pathogenic and non-pathogenic.

• It is unclear how the false negative rate was calculated, given the absence of follow-up data.

Additionally, there are several considerations that could be addressed to strengthen the findings:

Long-Term Follow-Up: Although some follow-up results have been described, the study does not seem to include long-term data on how the NIPT results have actually impacted neonatal outcomes or the health of children in the long term.

Lack of In-Depth Analysis on High-Risk Pregnancies: Data on outcomes for high-risk pregnancies, such as those with maternal morbidity or negative neonatal outcomes, are poorly addressed. These data could enhance the understanding of the usefulness of NIPT not only in normal pregnancies but also in high-risk situations.

6. PLOS authors have the option to publish the peer review history of their article (what does this mean? ). If published, this will include your full peer review and any attached files.

**Do you want your identity to be public for this peer review?** For information about this choice, including consent withdrawal, please see our Privacy Policy .

Reviewer #1: No

Reviewer #2: No

Reviewer #3: No

---

## [Author Response · Author response to Decision Letter 1]

27 Jun 2025

Dear Editor,

Manuscript Title: Clinical experience of genome-wide non-invasive prenatal testing as a first-tier screening test in a cohort of 59,771 pregnancies

Manuscript ID: PONE-D-25-18695 -[EMID:66cdbb8b69d2ebbf]

Thank you for your email regarding our submission. We sincerely appreciate the detailed suggestions provided by your esteemed journal. In response to the journal requirements and reviewers’ comments, we have carefully revised the manuscript and addressed all the points raised. Below, we provide a response to reviewers and we hope that these replies and revisions meet the journal’s standards and address the concerns raised.

Journal Requirements:

Requirement 1: Please ensure that your manuscript meets PLOS ONE's style requirements, including those for file naming.

Reply: We have carefully revised the manuscript and the file naming according to your journal's formatting guidelines to ensure full compliance with PLOS ONE's style requirements.

Requirement 2: This research was supported by 2024 High-quality Development Research Project of Shenzhen Bao'an Public Hospital (Grant number: BAGZL2024154), and National Natural Science Foundation of China (Grant number: 82203291).

Reply: The funders had no role in the research. As stated in the cover letter: "The funders had no role in study design, data collection and analysis, decision to publish, or preparation of the manuscript."

Requirement 3: Please remove all personal information, ensure that the data shared are in accordance with participant consent, and re-upload a fully anonymized data set.

Reply: All uploaded data were removed personal information and obtained participant consent. These protocols were stated in the "Ethics Statement" subsection of the original manuscript, as follows: "Written informed consent was obtained from individual participants or their guardians. Records were accessed on 20th March 2022 and subsequently anonymized to protect the personal privacy of all pregnant women.".

Requirement 4: Please include captions for your Supporting Information files at the end of your manuscript, and update any in-text citations to match accordingly.

Reply: The captions for the Supporting Information files have been added in the "Supporting Information" section at the end of the manuscript, and the in-text citations have been updated accordingly.

Reviewers' comments:

Reviewer #1

Comment 5: First, the manuscript does not report the test failure rate or the number of samples for which results were inconclusive or uninterpretable. This information is essential to assess the practical applicability and reliability of the test in a real-world clinical setting.

Reply: We retrospectively reviewed genome-wide non-invasive prenatal testing (GW-NIPT) results from November 2017 to May 2021, and found that a total of 106 cases failed due to fetal fraction below 3.5%. Thus, a total of 59,877 patients underwent testing, with 59,771 patients ultimately included in the analysis. In the manuscript, we made corresponding modifications in the "Subjects and Methods", "Results" sections and Figure 1.

(1)Subjects and Methods Section

In the "Study subjects" subsection, We change the first sentence into: "The retrospective study enrolled 59,877 singleton or twin pregnant women opting for GW-NIPT as a first-tier prenatal test from November 2017 to May 2021 at Shenzhen Baoan Women's and Children's Hospital in China. Among these, 59,771 women were included in the subsequent analysis due to 106 test failures."

(2) Results Section

We revised the heading of the first section to "Study population characteristics and GW-NIPT results", and added the following sentence described the test failure rate: "A total of 59,877 pregnancies who underwent GW-NIPT from November 2017 to May 2021 were accessioned for this study and 106 were tested failure (fetal fraction <3.5%). The test failure rate was 0.18% (106/59877) and 59,771 cases were enrolled in further analysis (Fig 1)."

(3) Figure 1 (See the file "Fig 1")

Fig 1. Flowchart of the GW-NIPT results and outcomes of pregnant women. GW-NIPT, genome-wide non-invasive prenatal test; n, number; T, trisomy; SCA, sex chromosome aneuploidy; RAA, rare autosomal aneuploidy; CNV, copy number variation; CMA, chromosomal microarray analysis; FISH, fluorescence in situ hybridization. *Include 1,315 cases that underwent both karyotyping and CMA.

Comment 6: Second, while the authors report performance metrics such as sensitivity and specificity, the methodology behind these calculations is not sufficiently explained. It is unclear how true positives were defined and whether confirmation was based on invasive diagnostic testing, postnatal outcomes, or clinical evaluation. A detailed description of the reference standard and classification criteria is necessary to validate the reported performance.

Reply: In terms of the definitions of true positives, false positives, etc., associated with the calculation of performance metrics such as sensitivity and specificity, we have modified Figure 1 (See the file "Fig 1" and the Comment 5) to make the NIPT results and follow-up process clearer and more visible, and offered a detailed explanation in the "Subjects and Methods" section.

In the subsection of "Invasive prenatal diagnosis and postnatal outcomes", we modified the first paragraph to "Pregnant women who required further examination would have amniocentesis for karyotyping, chromosomal microarray analysis (CMA), or fluorescence in situ hybridization (FISH), following the results of NIPT, ultrasound examination and other medical history. A small number of cases underwent diagnostic testing using chorionic villus sampling (CVS), umbilical cord blood, and fetal tissue." We aimed to explicitly state that invasive prenatal diagnosis included karyotyping, CMA, and FISH. These diagnostic tests can be selected individually or in combination by doctors and patients based on comprehensive evaluation.

We also added two following paragraphs in this section to describe the postnatal outcomes and classification criteria for the results:

"For postnatal outcomes, we included information on the types of delivery (live birth vs. stillbirth), the presence of any birth defects, and the clinical evaluation of birth defects. When postnatal outcomes was not recorded, potential reasons included termination of pregnancy (TOP) or delivery at another hospital, which we did not differentiate in this study."

"We classified NIPT results into true positive (TP), false positive (FP), true negative (TN) and false negative (FN) according to invasive testing results or the postnatal outcomes. The criteria for judgment of NIPT results are as follows: (1) T21, T18, T13 and RAA can be comfirmed via invasive testing (karyotyping, CMA or FISH) or postnatal outcomes due to their distinct clinical features, such as atelencephalia in common trisomies and cardiovascular defects in RAA [22, 23]. (2) SCA was comfirmed by invasive testing of karyotyping, CMA or FISH. (3) CNV (≥5 Mb) was comfirmed by invasive testing of CMA."

References:

22.Syngelaki A, Guerra L, Ceccacci I, Efeturk T, Nicolaides KH. Impact of holoprosencephaly, exomphalos, megacystis and increased nuchal translucency on first-trimester screening for chromosomal abnormalities. Ultrasound Obstet Gynecol. 2017;50(1):45-48.

23. Hu R, Huang W, Zhou W, et al. Phenotypic findings and pregnancy outcomes of fetal rare autosomal aneuploidies detected using chromosomal microarray analysis. Hum Genomics. 2022;16(1):64.

Comment 7: Additionally, although the study employs a genome-wide approach, the discussion could better emphasize the advantages of this strategy. Specifically, the authors should elaborate on how genome-wide NIPT may increase diagnostic yield by detecting atypical, rare, or subchromosomal CNVs that would not be identified by targeted assays. This is a key strength of the approach and should be clearly articulated.

Reply: We added a comparative analysis between GW-NIPT and targeted CMA in the final paragraph of the Discussion section:

"Based on targeted array CGH using Fetal DNA Chip Version 1.2, CMA can detect all common aneuploidy diseases and over 100 types of MMSs of known gene sequence, yet has limitations like missing some small chromosomal abnormalities, copy number polymorphisms, mosaicism, and rearrangements. GW-NIPT enables detection of a broader range of genomic variations and is capable of identifying atypical, rare, or subchromosomal CNVs that would not be identified by targeted assays. The Algorithms of GW-NIPT developed by BGI-Shenzhen were used to detect CNV and the reads were aligned to the human genome (hg19). After raw data processing, GC correction, intra-batch correction and Principal Component Analysis (PCA), Hidden Markov Model (HMM) was used to infer the hidden states of each window (normal, deletion and duplication) [40]. Concurrently, GW-NIPT identified 2 cases of SCA mosaicism (comfirmed by karyotyping) that were undetected by CMA. These findings underscore the clinical necessity of implementing GW-NIPT."

References:

40. Chen S, Zhang L, Gao J, et al. Expanding the Scope of Non-invasive Prenatal Testing to Detect Fetal Chromosomal Copy Number Variations. Front Mol Biosci. 2021;8:649169.

Comment 8: Importantly, the manuscript does not specify the resolution limit for CNV detection (i.e., the minimum size of CNVs that can be reliably identified). This technical parameter is critical for interpreting the scope and limitations of the assay and for comparing its performance with other technologies such as array-CGH. The authors should clearly indicate this resolution threshold and how it was validated.

Reply: This retrospective study evaluated the performance of GW-NIPT in a clinical setting. To minimize unnecessary invasive procedures, only CNVs ≥5 Mb were reported in the test results. For a more direct expression, we have changed it to "CNV (≥5 Mb)" in the entire manuscript.

There are several relevant articles supporting this resolution limit for CNV detection. R Li, et al. reported that among CMA-positive samples, the detection rate of NIPT was 90.9% for CNVs >5 Mb, compared to only 14.3% for CNVs <5 Mb. [41]. Furthermore, studies have reported that for CNVs <5 Mb detected by expanded NIPT, the positive predictive values (PPVs) were only approximately 20%, whereas for CNVs ≥5 Mb, the PPVs ranged from 30% to 70% [14, 19].

Additionally, although CMA testing in invasive prenatal diagnosis may report clinically significant microduplications or microdeletions greater than 100kb of genomic copy number, we restricted our comparison between NIPT and CMA to cases where NIPT was negative for CNVs but CMA was positive—including only those with CMA-detected CNVs ≥5 Mb.

References:

14. Xue H, Yu A, Lin M, et al. Efficiency of expanded noninvasive prenatal testing in the detection of fetal subchromosomal microdeletion and microduplication in a cohort of 31,256 single pregnancies. Sci Rep. 2022;12(1):19750.

19. Li C, Xiong M, Zhan Y, et al. Clinical Potential of Expanded Noninvasive Prenatal Testing for Detection of Aneuploidies and Microdeletion/Microduplication Syndromes. Mol Diagn Ther. 2023;27(6):769-779.

41. Li R, Wan J, Zhang Y, et al. Detection of fetal copy number variants by non-invasive prenatal testing for common aneuploidies. Ultrasound Obstet Gynecol. 2016;47(1):53-57.

Comment 9: Furthermore, a demographic summary of the study cohort—including maternal age, gestational age, and clinical indications for testing—would provide valuable context for the reader. The authors are also encouraged to include more detail on the bioinformatic workflow used for CNV calling, including the software tools, filtering criteria, and any validation methods applied. Sharing access to the analysis pipeline, when possible, would greatly enhance the study's transparency and reproducibility.

Reply: A demographic summary of the study cohort—including maternal age, body mass index (BMI), gestational age, and pregnancy types, was presented in Table 1 (See the original manuscript). We have made the following modifications in the "Results" section:

"Demographic characteristics of the study population were summarized in Table 1. The average maternal age was 29.5 (±4.1) years which is slightly higher than the national average age of pregnant women of 28.8 (±5.1) years from 2016 to 2020 [24], and 11.6% (6946/59771) of pregnant women were in the advanced maternal age (≥35). The mean body mass index (BMI) was 21.7 (±3.0) kg/m2 and the median gestational age was 17 weeks (range: 12-39.3)."

Additionally, the bioinformatics software used for analyzing CNVs and other chromosomal abnormalities was BGI HALOS, as mentioned in the “Laboratory analysis of GW-NIPT” of the “Subjects and Methods” section. We added following description of the detailed bioinformatics workflow used for CNV calling in the last paragraph of the "Discussion" section:

"GW-NIPT enables detection of a broader range of genomic variations and is capable of identifying atypical, rare, or subchromosomal CNVs that would not be identified by targeted assays. The Algorithms of GW-NIPT developed by BGI-Shenzhen were used to detect CNV and the reads were aligned to the human genome (hg19). After raw data processing, GC correction, intra-batch correction and Principal Component Analysis (PCA), Hidden Markov Model (HMM) was used to infer the hidden states of each window (normal, deletion and duplication) [40]."

References:

24. Zhou Y, Yu H, Wang A, et al. Temporal and geographic trends in maternal and paternal ages in China from 2016 to 2020. Chinese Journal of Reproductive Health. 2024;35(4):301-305, 312.

40. Chen S, Zhang L, Gao J, et al. Expanding the Scope of Non-invasive Prenatal Testing to Detect Fetal Chromosomal Copy Number Variations. Front Mol Biosci. 2021;8:649169.

Comment 10: Finally, some minor editorial improvements could enhance the clarity and flow of the text, particularly in the abstract and results sections.

Reply: We have made several revisions to enhance the clarity and flow of the text, particularly in the abstract and results sections.

(1) Abstract Section

We replaced "NIPT" with "GW-NIPT" to explicitly denote genome-wide non-invasive prenatal testing. In the subsection of results, Our modifications are as follows: "Among 59,877 pregnant women who underwent GW-NIPT, 59,771 were successfully tested. Of these, 499 (0.83%) were identified with 504 high-risk fetal chromosomal abnormalities, including 5 cases each carrying two distinct abnormalities. Follow-up analysis demonstrated that GW-NIPT sensitivity exceeded 97% for fetal aneuploidies and was 63.6% for CNV (≥5 Mb). The positive predictive values for T21, T18, T13, sex chromosome aneuploidy, rare autosomal aneuploidy, and CNV (≥5 Mb) were calculated as 83.1%, 25.8%, 10.3%, 51.9%, 2.0%, and 33.9%, respectively."

(2) Results Section

The "Study population characteristics and GW-NIPT results" subsection presents the illustration of the overall NIPT results and diagnostic process, as follows:

"A total of 59,877 pregnancies who underwent GW-NIPT from November 2017 to May 2021 were accessioned for this study and 106 were tested failure (fetal fraction <3.5%). The test failure rate was 0.18% (106/59877) and 59,771 cases were enrolled in further analysis (Fig 1). In the results of GW-NIPT, 499 (0.8%) patients were suspected to be high risk, of which 5 carried two types of abnormal high risk. Among 504 high-risk cases, 367 cases were confirmed by follow-up diagnosis, and 137 cases were lost to follow-up. 59,272 cases showed low-risk results for all of the detected chromosomal abnormalities, of which 1,402 cases underwent invasive CM

---

## [Editor Report · Decision Letter 1]

Clinical experience of genome-wide non-invasive prenatal testing as a first-tier screening test in a cohort of 59,771 pregnancies

PONE-D-25-18695R1

Dear Dr. Jianxin Zhen,

We’re pleased to inform you that your manuscript has been judged scientifically suitable for publication and will be formally accepted for publication once it meets all outstanding technical requirements.

Kind regards,

Giuseppe Novelli

Academic Editor

PLOS ONE
---

## [Editor Report · Acceptance letter]

PONE-D-25-18695R1

PLOS ONE

Dear Dr. Zhen,

I'm pleased to inform you that your manuscript has been deemed suitable for publication in PLOS ONE. Congratulations! Your manuscript is now being handed over to our production team.

Kind regards,

on behalf of

Prof. Giuseppe Novelli

Academic Editor

PLOS ONE